**Comparing soil moisture anomalies from multiple independent sources over different regions across the globe**

Carmelo Cammalleri, Jürgen V. Vogt, Bernard Bisselink, Ad de Roo

European Commission, Joint Research Centre (JRC), Ispra, Italy.

*Correspondence to:* C. Cammalleri, European Commission - Joint Research Centre, via E. Fermi 2749, I-21027 Ispra (VA), Italy. Bldg. 26b, Room 140, TP 267. Phone: +39 (0)332.78.9869, e-mail: carmelo.cammalleri@ec.europa.eu.

**Abstract:** Agricultural drought events can affect large regions across the World, implying the urge for a suitable global tool for an accurate monitoring of this phenomenon. Soil moisture anomalies are considered a good metric to capture the occurrence of agricultural drought events, and they have become an important component of several operational drought monitoring systems. In the framework of the JRC Global Drought Observatory (GDO, http://edo.jrc.ec.europa.eu/gdo/) the suitability of three datasets as possible representation of root zone soil moisture anomalies has been evaluated: (1) the soil moisture from the Lisflood distributed hydrological model (namely LIS), (2) the remotely sensed Land Surface Temperature data from the MODIS satellite (namely LST), and (3) the ESA Climate Change Initiative combined passive/active microwave skin soil moisture dataset (namely CCI). Due to the independency of these three datasets, the Triple Collocation (TC) technique has been applied, aiming at quantifying the likely error associated to each dataset in comparison to the unknown true status of the system. TC analysis was performed on five macro-regions (namely North America, Europe, India, Southern Africa and Australia) detected as suitable for the experiment, providing insight into the mutual relationship between these datasets as well as an assessment of the accuracy of each method. Even if no definitive statement on the spatial distribution of errors can be provided, a clear outcome of the TC analysis is the good performance of the remote sensing datasets, especially CCI, over dry regions such as Australia and Southern Africa, whereas the outputs of LIS seem to be more reliable over areas that

are well monitored through meteorological ground station networks, such as North America and
Europe. In a global drought monitoring system, the results of the error analysis are used to design a
weighted-average ensemble system that exploits the advantages of each dataset.
**1.  Introduction**
Drought is a recurring natural extreme, triggered by lower than normal rainfall, often exacerbated
by a strong evaporative demand due to high temperatures and strong winds. Drought events may occur
in all climates and in most parts of the world, since drought is defined as a temporary deviation from the
local normal condition. Due to the usually wide extension of the interested area, drought affects millions
of people across the Globe each year (Wilhite, 2000).
On the basis of the economic and natural sectors impacted by this phenomenon, a drought event is
usually classified in meteorological, agricultural and hydrological drought, depending on the persistence
of the water deficit within the hydrological cycle. Of particular interest for this study are the agricultural
(or ecosystem) drought events, defined as prolonged periods with drier than usual soils that negatively
affect vegetation growth and crop production, and, as a consequence, human welfare (Dai, 2011).
Soil moisture is commonly seen as one of the most suitable variables for monitoring and
quantifying the impact of water shortage on vegetated lands due to its effects on the terrestrial biosphere
and the feedbacks into the atmospheric system. As a consequence, time-aggregated soil moisture
anomalies (e.g., monthly) are included in numerous drought monitoring systems from regional to
continental scales (i.e., European Drought Observatory, http://edo.jrc.ec.europa.eu; United States
Drought    Monitor,    http://droughtmonitor.unl.edu;    African    Flood    and    Drought    Monitor,
http://hydrology.princeton.edu/adm/; among others).
In the context of drought monitoring, the soil moisture dynamic over large areas is usually
modelled through either distributed hydrological models or land-surface schemes of climate models
(Crow et al., 2012; Sheffield et al., 2004), as well as by thermal or passive/active microwave remote
sensing-derived quantities (see e.g., Anderson et al., 2007; Houborg et al., 2012; Mo et al., 2010). With

regard to a global-scale monitoring, remote sensing-based approaches have the advantage of an intrinsic

worldwide coverage. However, microwave sensors, can explore only the first few centimeters of soil

and are characterized by a decreasing sensitivity with increasing vegetation coverage (Jackson, 2006).

In the case of thermal data, the lack of coverage during cloudy conditions and the nontrivial connection

between thermal and soil moisture signals (Price, 1980) are other limitations. On the contrary,

diagnostic models allow for a continuous monitoring of soil moisture at the desired soil depths, but the

accuracy of the data is constrained by uncertainties in the parameterization of soil hydrological

characteristics, as well as by the actual availability of near-real time reliable meteorological forcing

data. Generally, the use of in-situ observations for large area monitoring is limited, mainly due to the

lack of long records, the sparseness of recording stations and the high spatial heterogeneity of soil

moisture fields.

It follows that both satellite measurements and model predictions are subject to errors and

uncertainties that need to be accounted for in their interpretation and application (Gruber et al., 2016).

This also suggests that a monitoring system based on a single dataset is rarely capable of providing

global reliable estimates, and a combination of different data sources is desirable in order to minimize

the errors in the detection of drought events. Recently, Cammalleri et al. (2015) demonstrated the value

of an ensemble of modelled soil moisture anomalies for drought monitoring over Europe, similarly to

the findings of the U.S. National Land Data Assimilation System (NLDAS) (Dirmeyer et al., 2006).

However, a key point in combining different modelled data is the need to estimate the affinity and

divergence between the models across the modelling domain.

In the most recent years, the Triple Collocation (TC) technique (Stoffelen, 1998) has been

established as a practical approach to evaluate the unknown error variance (with respect to the truth) of

three mutually independent measurement systems without knowing the "true" status of the system

(Yilmaz and Crow, 2014). This technique has been widely applied in hydrology to estimate errors in

soil moisture, as well as to evaluate precipitation and vegetation property indicators (Dorigo et al.,

2010; McColl et al., 2014). One key requirement in TC is the existence of linearity between the three

estimates and the truth, which can fail in the case of strongly seasonal geophysical variables such as soil

moisture (Su et al., 2014). Luckily, drought monitoring systems are usually based on soil moisture anomalies rather than actual values, hence providing a partial remedy to this problem and making soil moisture anomalies directly suitable for this methodology (Miralles et al., 2010). However, since most of TC studies focused on soil moisture dynamics rather than standardized anomalies, specific analyses are required to evaluate the accuracy of each dataset across the spatial domain.

In the frame of an operational monitoring of agriculture and ecosystem drought, the availability of soil moisture, or proxy datasets available in near-real time, is crucial; within the Global Drought Observatory (GDO, http://edo.jrc.ec.europa.eu/gdo/), developed by the Joint Research Centre (JRC) of the European Commission, the soil moisture outputs of the Lisflood hydrological model and the Land Surface Temperature (LST) anomalies derived from the Moderate-Resolution Imaging Spectroradiometer (MODIS) onboard the Terra satellite have been detected as suitable datasets for a near-real time monitoring. In particular, Cammalleri and Vogt (2016) have highlighted how monthly-average LST anomalies represent the best proxy of soil moisture variations across different climates in Europe when compared to other LST-derived quantities.

As a third dataset for the TC analysis, the combined active/passive microwave soil moisture dataset produced by the European Space Agency (ESA) in the context of the Climate Change Initiative (CCI) is used; even if this dataset is not currently updated in near-real time, it represents a valuable reference dataset for a global consistent time-series of microwave-based soil moisture maps (also, near-real time updating is foreseen in the framework of the Copernicus Climate Change Services).

The agreement between anomaly time-series derived from these three products has not been fully investigated in the literature, especially at global scale; hence, given the independency of the three sources of data (hydrological model, thermal and microwave remote sensing) and the likely fulfilling of the main TC key hypothesis (i.e., independency between the errors of the three datasets), the TC approach seems suitable for quantifying the spatial distribution of the errors associated to each dataset.

Following these considerations, the overall goal of this study is twofold. First, the agreement between the monthly anomalies of the three datasets is evaluated, in order to identify the macro-areas where a reliable monitoring of soil moisture extreme conditions can be performed based on these three

datasets that are available globally and suitable for use in a near-real time monitoring system. Second,
the TC analysis is performed over those macro-areas in order to quantify the spatial distribution of the
expected random errors for each model compared to the unknown true status. The ultimate objective of
the error analysis reported in this study is to provide information on the accuracy of the datasets that can
be injected into a weighted-average ensemble procedure for a near-real time detection of the occurrence
of ecosystem drought events, thus contributing to the future development of a robust agricultural
drought monitoring index within the GDO system.

**2.    Methods**

Drought events are commonly defined as prolonged periods during which a given drought
indicator significantly deviates from the usual condition for the specific site and period (e.g., soil
moisture content is lower than the climatology). Following this definition, this study will focus on
standardized z-score values in order to make the different datasets directly comparable (i.e., minimizing
the differences related to seasonality, soil depth, etc.). Specifically, monthly z-score values, or
anomalies, are evaluated as:
$$Z_{x,i,k} = \frac{x_{i,k} - \mu_{x,i}}{\sigma_{x,i}} \tag{1}$$
where $x_{i,k}$ is the monthly average variable for the $i$-th month at the $k$-th year, $\mu_{x,i}$ and $\sigma_{x,i}$ are the long-
term average and standard deviation of the variable $x$ for the $i$-th month, respectively. The baseline
period adopted to compute the twelve $\mu$ and $\sigma$ monthly reference values should be of 15-30 years in
order to ensure a stable benchmark. The three datasets used here, as described in the next section, are
the root zone soil moisture data from the Lisflood model ($x$ = LIS), the ESA Climate Change Initiative
skin soil moisture microwave combined product ($x$ = CCI) and the thermal remote sensing derived Land
Surface Temperature ($x$ = LST); in the case of LST data, the sign of the anomalies is reversed due to the
expected inverse relationship between soil moisture and LST.
The monthly aggregation period is chosen to ensure a statistical robustness of the computed
anomalies, as well as to minimize the presence of missing data in the remote sensing datasets due to
sub-optimal acquisition conditions (e.g., cloudy days for LST). The transition from daily data to
monthly aggregated values also ensures a reduction in the likely discrepancies among the three datasets
introduced by the differences in the explored soil depth, since the phase shift in time-aggregated
quantities is usually less marked (Campbell and Norman, 1998). Additionally, the anomalies computed
according to Eq. (1), characterized by a null average and a unitary standard deviation, allow for a direct
comparison of the different datasets thanks to the removal of potential biases. In the particular case of a
regression analysis between two standardized anomaly quantities, the Pearson correlation coefficient, $R$,
represents not only a measure of the linear dependency of the two random variables but also the slope of
the linear relationship and a proxy of the difference and biases of the two datasets. In this respect, $R$ can
be seen as a good synthetic descriptor of the relationship between two standardized z-score datasets.
The statistical significance of the existence of a positive correlation can be evaluated by means of the
Student's t-test (2 sided) by computing the $R$ value corresponding to a significance level $p = 0.05$.
Analysis of the correlation among the datasets is interesting in the framework of the triple
collocation (TC) technique and its basic hypotheses. In TC, a first key hypothesis is the existence of
linearity between the 'true' status of the system and the three models; this is formally expressed as:
$$z_x = \alpha_x + \beta_x z_\Theta + \varepsilon_x \qquad\qquad (2)$$
where $z_\Theta$ is the unknown true dataset of soil moisture anomalies, $\alpha_x$ and $\beta x$ are the systematic slope and
bias parameters for the dataset $x$ with respect to the truth, and $\varepsilon_x$ is the additive zero-mean random noise.
It follows that the absence of a statistically significant linear relation between all three models openly
violates this hypothesis.
Other key underling hypotheses of TC are the stationarity of both signals and errors, the
independency between the errors and the signal (error orthogonality) and the independence between the
errors of the three datasets (zero-cross correlation) (Gruber et al., 2016). Finally, operational limitations
regard the minimum sample size of each dataset, which is commonly assumed equal to 100 values
(Scipal et al., 2008; Dorigo et al., 2010), even if some other authors suggest much larger sample sizes
for a lower relative uncertainty (Zwieback et al., 2012).
Under these assumptions, Stoffelen (1998) proposed a formulation to estimate each model error
variance, $\sigma^2{}_{\varepsilon x}$, based on a combination of the covariance between the datasets. In this approach, known
as the covariance notation (Gruber et al., 2016), the error variance values are computed without a
common (arbitrary) reference dataset as:

$$
\begin{aligned}
\sigma_{\varepsilon_1}^2 &= \sigma_1^2 - \frac{\sigma_{12}\sigma_{13}}{\sigma_{23}} \\
\sigma_{\varepsilon_2}^2 &= \sigma_2^2 - \frac{\sigma_{21}\sigma_{23}}{\sigma_{13}} \\
\sigma_{\varepsilon_3}^2 &= \sigma_3^2 - \frac{\sigma_{31}\sigma_{32}}{\sigma_{12}}
\end{aligned}
\tag{3}
$$


where, for the sake of simplicity, LIS, LST and CCI were renamed 1, 2, 3, respectively. The first term
on the right side of Eqs. (3) represents the single model data variance, whereas the second term
represents the so-called sensitivity of the model to variations in the true status, which is a function of the
covariance terms between the three models. The advantage of this formulation is to directly estimate the
unscaled error variances, which can (eventually) be scaled to a common data space, if needed.
In the case of the application of the covariance notation to standardized quantities (with zero mean
and unitary standard deviation), the error variance values computed through Eqs. (3) are expressed as
dimensionless multiples of standard deviation, and a transformation to a common data space is not
needed.
Different performance metrics can be derived from the covariance notation, including relative
error variance metrics such as the fractional root-mean-squared-error (fRMSE, Draper et al., 2013) and
the correlation coefficient of each model with the underlying true signal (McColl et al., 2014).
However, these metrics can be derived from each other by means of simple relationships (see Gruber et
al., 2016) and they are analogous to the absolute error variance values in the case of z-scores that have
known unitary dataset variance.

**3.   Data and Materials**

### 3.1 Lisflood model soil moisture

Root zone soil moisture dynamics are simulated by means of the Lisflood model (de Roo et al., 2000), a GIS-based distributed hydrological rainfall-runoff-routing model designed to reproduce the main hydrological processes that occur in large and trans-national European river catchments. The model simulates all the main hydrological processes occurring in the land-atmosphere system, including infiltration, actual evapotranspiration, soil water redistribution in three sub-layers (surface, root zone and sub-soil), surface runoff rooting to channel, and groundwater storage and transport (Burek et al., 2013).

Static maps used by the model are related to topography (i.e., digital elevation model, local drain direction, slope gradient, elevation range), land use (i.e., land use classes, forest fraction, fraction of urban area), soil (i.e., soil texture classes, soil depth), and channel geometry (i.e., channel gradient, Manning's roughness, bankfull channel depth, channel length, bottom width and side slope). Root zone depth is defined for each modelling cell on the basis of soil type and land use, where the soil-related hydraulic properties are obtained from the ISRIC 1-km SoilGrids database (Hengl et al., 2014), whereas topography data are obtained from the Hydrosheds database (Lehner et al., 2008).

Daily meteorological forcing maps are derived from the European Centre for Medium-range Weather Forecasts (ECMWF) data as spatially resampled and harmonized by the JRC Monitoring Agricultural ResourceS (MARS) group. The dataset includes daily average air temperature, potential evapotranspiration (for soil, water and reference surfaces) and total rainfall at 0.25 degree spatial resolution, which were resampled on the model grid using the nearest neighbour algorithm.

The model run used in this study includes daily maps at 0.1 degree resolution between 1989 and 2015; the grid domain of this dataset is used as reference for the other two, whereas the baseline for the anomalies computation is defined by the period 2001-2015 in order to match the LST data availability. Monthly data to be used in Eq. (1) are computed as a simple average of all the data available for each month, given that no gaps can be found in this dataset due to its continuous nature as hydrological

model. However, some areas where masked out due to the minimum or null temporal dynamic of soil
moisture, such are Greenland and the Sahara desert.

*3.2 Land Surface Temperature dataset*

The use of the Land Surface Temperature (LST) anomalies as a proxy of soil moisture anomalies
is based on the well-known role of LST in the surface energy budget as a control factor for the
partitioning between latent and sensible heat fluxes. In recent years, the existence of a connection
between soil moisture and LST has been analyzed, mainly through the thermal inertia and the triangle
methods (e.g., Carlson 2007; Verstraeten et al., 2006), as well as by using LST as a direct proxy (see
e.g., Park et al., 2014; Srivastava et al., 2016). In a study over the pan-European domain, Cammalleri
and Vogt (2016) have demonstrated the good agreement between monthly LST and LIS-based root zone
soil moisture z-score values during summer time, where LST outperforms other LST-based indicators
such as the day-night difference and the surface-air gradient.
Following these findings, this study adopts the dataset collected by the Moderate-Resolution
Imaging   Spectroradiometer   (MODIS)   sensor   on   board   of   the   Terra   satellite
(http://terra.nasa.gov/about/terra-instruments/modis) as a source of monthly-scale long records of LST
maps. In particular, the MOD11C3 monthly CMG (Climate Modelling Grid) LST product is used in this
study, which is constituted by monthly composited and averaged temperature and emissivity maps at a
spatial resolution of 0.05 degrees over a regular latitude/longitude grid; data for the period 2001–2015
are used, being the only fully completed years at the time of the analysis.
This monthly composite product is obtained as an average of the clear-sky data in the MOD11C1
products on the calendar days of the specific month, which are derived after re-projecting and re-
sampling of the MOD11B1 product. Details on the algorithm used to obtain the daily MOD11B1 maps
can be found in Wan et al. (2002); in summary, a double screening procedure is applied, based on: i) the
difference between the two independent LST estimates of the day/night algorithm (Wan and Li, 1997)

and the generalized split-window algorithm (Wan and Dozier, 1996), and ii) the histogram of the difference between daytime and nighttime LSTs.

LST monthly maps were spatially co-registered to the Lisflood 0.1 degree regular latitude/longitude grid by means of a simple average of the values within each cell, and anomaly maps were computed according to Eq. (1) by using only the data for which LST > 1 °C; this threshold value (commonly used in snowmelt and snow/rainfall discrimination procedures; WMO, 1986) allows removing from the analysis the data that are likely affected by snow/frost.

### 3.3 Microwave combined dataset

The ESA Climate Change Initiative (CCI) aims at developing a multi-satellite soil moisture dataset by combining data collected in both past and present by passive and active microwave instruments (Liu et al., 2012; Wagner et al., 2012). The current version of the dataset (v03.2) combines data from nine different sensors (SMMR, ERS-1/2, TMI, SSM/I, AMSR-E, ASCAT, WindSat, AMSR2 and SMOS) between 1978 and 2015.

Satellite-based microwave estimates of soil moisture are usually related to the first few centimeters of soil column (i.e., skin layer), which is quite closely related to the soil moisture content in the root zone (Paulik et al., 2014), except for very dry conditions in sandy soils. Additionally, numerous validations against land surface models have highlighted a good performance across the globe, with notable exceptions over densely vegetated areas (e.g., Loew et al., 2013).

The algorithm adopted to merge the different data sources is the one developed by Liu et al. (2012), which is a three-step procedure that: i) merges the original passive microwave products, ii) merges the original active microwave products, and iii) blends the two merged products into a single final dataset. The merging procedure of passive datasets includes pixel-scale separation between seasonality and anomalies, rescaling of the data based on the piece-wise cumulative distribution function (CDF) and merging of the dataset using a common reference seasonality. For the active microwave instruments, the CDFs are directly used to rescale the data under the assumption that active

datasets have an identical dynamic range, this mainly due to the limited overlap between datasets. The
final blending of the two merged datasets is obtained by adopting a common resolution of
approximately 25 km and daily frequency, as well as by using the GLDAS-1-Noah model
(ftp://agdisc.gsfc.nasa.gov/data/s4pa/) as a reference dataset for the CDF matching.
In this study, the daily blended dataset is spatially resampled to a 0.1 degree regular
latitude/longitude grid (the same used in Lisflood simulations) by means of the nearest neighbor
algorithm, and successively aggregated to monthly time scale by simply averaging the data (only if at
least 8 daily values were available in the specific month). Monthly average maps were converted into z-
score maps by using the baseline period 2001-2015 (the timeframe available for the LST dataset).
Monthly aggregated z-score values of skin soil moisture are analyzed, jointly with the other two
datasets, under the assumption that time-aggregation and normalization procedures minimize some of
the discrepancies that are likely present between skin and root zone daily time-series.

**4.   Results and Discussion**

4.1 Linear regression analysis

Considering the assumption of linearity between each one of the datasets and the unknown true
status of the system in TC, a preliminary analysis on the linear correlation between the three anomaly
products has been performed in order to detect the macro-areas where the TC procedure can be applied
without violating this basic hypothesis. The correlation analysis was performed by using only the
monthly anomalies that were available for all three datasets, with at least a sample size of 100 values
(max sample size = 12 months × 15 years = 180), and by defining a minimum correlation threshold
($R_{0.05}$) that ensures a statistical significance of the linear relationship on the basis of the Student's t-test
(at $p = 0.05$).
The map in Fig. 1 reports in grey the areas where all three datasets are significantly linearly
correlated according to the described criteria, representing the areas where the first basic hypothesis of

the TC is not clearly violated. It is worth to point out that some areas are excluded from the analysis by the lack of data in LIS (low temporal variability, as over Greenland and the Sahara desert), LST (due to the minimum temperature threshold or low temporal variability) or CCI (densely vegetated areas, such as the Amazon forest and the Congo basin). These results suggest to focus the successive detailed analysis on five macro-regions (demarked by the boxes in Fig. 1) that have consistent positive correlation values for all the three datasets; these areas are named, from now on, as: 1) NA (North America, including the contiguous U.S. and Mexico), 2) EU (Southern and Central Europe), 3) SA (Southern countries of the African continent and Madagascar), 4) IN (Indian subcontinent), and 5) AU (Australia)[*].

The correlation coefficient maps over those regions, obtained by inter-comparing the three datasets, are reported in Figs. 2 to 4, where the cells in red and yellow are the ones with negative or not-significant correlation, respectively, whereas the blue scale represents the cells with increasing significant linear correlation (from light to dark tones). The comparison between LIS and LST (Fig. 2) shows an overall good agreement between the two datasets, with only minor areas characterized by negative/not-significant correlation values; notably, low correlation values can be observed over the Great Lakes and Rocky mountain areas in the U.S., over the Alps in Europe, North Angola and Western Himalaya. Similar results can be observed in Fig. 3, where LIS and CCI datasets are compared; this comparison shows an increasing number of negative values in the Western U.S., the Alps, and Southern Turkey, but overall high correlation values across most of the five regions. Finally, the comparison between LST and CCI reported in Fig. 4 shows an increase of areas with low/not-significant correlation in the Eastern and Western U.S. and both North- and South-Eastern Europe and the Alps, whereas high correlation values can be observed all over the other regions.

On average, the data in Table 1 summarize the results obtained for all the regions together, as well as for each region independently, showing how CCI and LST are the two datasets best correlated to each other overall, even if this result is mainly driven by the results over the AU, SA and IN macro-areas. The LIS model data are similarly correlated to the ones of LST and CCI, with a more uniform

---

[*] Consider the countries and boundaries reported here only as indicative of the interested areas, and they may not in any circumstances be regarded as stating an official position of the European Commission.

distribution of the results across the various sub-regions. Another outcome of this analysis is that the area with the lowest average correlation between the three datasets is the EU, probably due to the high heterogeneity of this region at the 0.1 degree spatial scale.

Some of the discrepancies observed in Figs. 2 to 4 can be explained by the differences in both horizontal and vertical resolution of the three raw datasets. LIS is characterized by an higher spatial resolution (5-km) compared to CCI (25-km) and a vertical resolution that encompasses the full root zone against the skin soil moisture of the latter; LST has a spatial resolution close to LIS but a vertical resolution that varies as function of the vegetation coverage between skin (for bare soil) to root zone (for full vegetation coverage). The impact of such differences is partially reflected in the observed results, with CCI-LST better related over shallow soil in homogeneous areas, and LIS-LST better in agreement over sparse agricultural areas in Europe. Overall, it seems that the adopted expedients (i.e., monthly average, standardization) successfully minimized these issues, given that the results in Table 1 show a substantial and similar agreement of the three datasets in the main areas.

Additionally, the obtained results seem to suggest that it is reliable to adopt LST anomalies as proxy of soil moisture anomalies, since there is a clear consistency of LST anomalies with the other two datasets. Similar results were obtained by Fang et al. (2016) over the continental United States, where the outputs of the thermal-based ALEXI (Atmosphere Land EXchange Inverse) model compare well with soil moisture anomalies from CCI and the Noah land-surface model. This consideration allows applying the TC analysis to the LST dataset as well, whereas most of the studies in the literature focus on land surface modelled and microwave soil moisture datasets (i.e., Dorigo et al., 2010; Gruber et al., 2016; Su et al., 2014) with only few notable exceptions including thermal data (e.g., Hain et al., 2011; Yilmaz et al., 2012).

4.2 Triple collocation analysis

The outcomes of the correlation analysis were used to detect the cells suitable for the TC technique; since a key hypothesis of the technique is the existence of a linear relation between each

model and the (unknown) truth, a necessary condition (even if not sufficient) is the existence of linear relationships among the three datasets. As outcome of the correlation analysis, around 10% of the five macro-areas were removed from the TC analysis due to the absence of this basic condition.

The maps in Figs. 5 to 7 show the main outcome of the TC analysis, which is the spatial distribution of the error variance (dimensionless, showing a multiple of the model standard deviation) for each model, as detailed by Eqs. (3). The blank areas in those maps correspond to the cells where no significant linear correlation was observed between all three datasets. The results for LIS (Fig. 5) show that the highest errors are observed over the Western U.S., Northern Cape in South Africa and Western/Southern Australia, whereas the lowest errors are observed over the Eastern U.S. On the opposite, the LST dataset displays the highest errors over the latter area (Fig. 6), whereas the lowest errors are observed over Queensland in Australia, Eastern Cape in South Africa and Lesotho. The maps in Fig. 7 show that the CCI dataset has consistent patterns of low error variance values over most of Australia, Western India and Central U.S.

Overall, on the one hand, it seems evident how CCI tends to outperform the other two methods over dry areas such as Australia and South Africa, but on the other hand, a region like the U.S. is almost equally subdivided among the three datasets, where LIS performs better in the East, LST in the West and CCI in the center. Differences among products can be partially explained by the differences in the soil layer monitored by each dataset, i.e., the microwave system captures the skin soil moisture whereas Lisflood models the full root zone; indeed, even if the use of monthly anomalies allows minimizing some of the discrepancies, skin soil moisture remains more reliable for dry/bare areas (Das et al., 2015). Even if these considerations partially explain the agreement/disagreement of the three datasets, it is not straightforward to pinpoint in detail climate and/or vegetation derived patterns in the spatial distribution of the TC outputs.

These findings are summarized in the data reported in Table 2, where the average error variance for each model and macro-area is reported aside its spatial standard deviation. The data in Table 2 confirm that CCI has an overall better performance (lower errors) than LIS and LST, which perform quite closely, mainly thanks to the very low error variance observed over Australia and, to a minor

extend, Southern Africa. The LIS model performs better over NA and EU regions, likely due to the better meteorological forcing datasets available over those regions compared to the other macro-areas (due to denser ground networks). The LST dataset seems to perform moderately well over all five macro-regions, with the only notable exception of EU; however, it rarely outperforms the other two datasets, constituting a "second-best" option in most of the cases. It is also worth to point out that the CCI dataset is often masked-out over those regions where the error of microwave techniques are likely high, whereas the data of the other two datasets are mostly produced globally; hence, a possible explanation of the better performance of CCI compared to LIS and LST may be linked to this preliminary screening of the data.

The outcome that LIS slightly outperforms the other two datasets over NA is in agreement with the results reported by Hain et al. (2011), where the Noah land-surface model slightly outperforms (on average) the microwave and thermal datasets over the contiguous U.S. However, it should be pointed out how the spatial distribution of the error estimates for LIS differs from the ones reported for Noah, likely due to the differences in both meteorological forcing and modelling approaches. Some qualitative analogies can also be observed with the results reported in Pierdicca et al. (2015), which show smaller average errors at daily time scale over Europe for the ERA-LAND modelled datasets compared to two microwave-based datasets, even if both the temporal scale and the adopted methodology of the latter differ from the ones used in our study. These previous studies seem to confirm that land modelling approaches are more reliable, on average, over these regions, likely due to the reliability of meteorological forcing and model parameterizations, even if there can be significant differences among the performances of different land-surface models.

Over the AU sub-region, the spatial distribution of the errors in CCI are quite in agreement with the results reported in Su et al. (2014) for two microwave datasets, with larger errors along the South-East Australian coast. This result supports the assumption that microwave data are more reliable over dry bare soil areas, which is further highlighted by the results obtained in SA and IN sub-regions. The subdivision of the NA domain in three main regions is similar to the one observed by Gruber et al. (2016) in comparing ASCAT and AMSR-E microwave datasets, suggesting key differences in the soil

moisture behavior over these three sub-regions. Overall, the spatial patterns of microwave and land
model errors show similarities with the ones observed by Dorigo et al. (2010), even if no thermal data
were included in their analysis.

The error variance values can also be interpreted as the correlation coefficient of each dataset with

the underlying true signal, following the definition of McColl et al. (2014). In fact, for the special case
of anomalies with unitary variance ($\sigma^2_x = 1$), the TC-derived $R_x$ of each dataset is simply equal to
$\sqrt{1 - \sigma^2_{\varepsilon_x}}$, which ranges on average over all five regions (not shown) between 0.91 (for CCI in AU) to
0.66 (for LST over EU); these values show a good capability of the datasets to capture, on average,
temporal variations in soil moisture anomalies.

4.3 Insights for a weighted-average ensemble procedure

In order to provide a simple synthetic representation of the likely best model for each area, the

maps in Fig. 8 depict for each cell the dataset with the lowest error variance by associating different
colors to the three datasets (red for LIS, blue for LST and green for CCI). Even if this approach is rather
simplistic, as it cannot account for two products performing really close over some areas, the major
relevant features, like the predominance of the CCI model over Australia, are made evident by these
maps.

The maps in Fig. 8 confirm CCI as the dataset with the lowest error variance values over most of

AU, SA and IN, whereas the three datasets almost equally split the other two macro-areas; this is even
more evident in the data reported in Table 3, where the percentage of sub-areas where each model is the
best is reported. These data confirm the good performance of CCI over AU, SA and IN macro-regions,
whereas the NA territory is almost equally divided among the three datasets and LIS outperforms both
LST and CCI over 50% of EU domain. In the latter, the areas where LIS dataset outperforms the other
two datasets partially resemble the results obtained by Pierdicca et al. (2011) for the ERA-LAND
model; however, the present study includes also remote sensing thermal data and not only microwave-
derived datasets. Overall, the CCI dataset outperforms the other two datasets in about 50% of the cells,
with the remaining almost equally split between LIS and LST.

Finally, the spatial distribution of the weighting factor of each dataset, computed according to the

least square theory (Yilmaz et al., 2012), is represented in Figs. 9 to 11. The color scale of the figures
was designed to represent in a neutral color the cells that have a weighing factor close to the one for a
simple-average (1/3), in green scale the weights greater than a simple-average (larger contribution) and
in orange the weights lower than the simple-average (smaller contribution). The visual intercomparison
of these three maps further emphasizes the good performance of the CCI product over AU and SA, the
best performance of LIS over the Eastern US and EU, and the good results obtained for LST in Western
US and Northern AU. It is worth noting that the use of a weighted average based on the TC error
analysis does not seem to bring advantages over large areas of central US, EU and Eastern IN where the
weighting factors are close to the ones for a simple arithmetic average. The behavior of the weighting
factors over the five macro-areas can be synthetized by the frequency diagram in Fig. 12. This plot
shows the high fraction of weighting factors > 0.4 for the CCI dataset, representing a predominant
contribution on the ensemble mean of this product over the others, whereas LST has a peak of
frequency center around 1/3 (arithmetic average) and LIS has a hint of a bi-modal distribution. These
data, together with the maps in Fig. 8, confirm the fact that CCI outperforms the other two datasets in
50% of the domain, whereas LST is often the second-best option behind either CCI or LIS.

**5.   Summary and Conclusions**

Three datasets have been compared as proxy of the unknown true status of soil moisture

anomalies in the context of a global drought monitoring system under development by the JRC of the
European Commission. Key assumption of the study is the inability of a single dataset to accurately
capture the soil moisture dynamic over the large range of variability of conditions that can be observed
at continental to global scale.
The inter-comparison between the three datasets, namely the outputs of the Lisflood hydrological
model (LIS), the MODIS-based land surface temperature (LST) and the combined active/passive
satellite microwave (CCI) data, confirms some inconsistencies between the three datasets over certain
areas, as well as the difficulties in comparing the three datasets over specific areas (e.g., Sahara desert,
Amazon rainforest) that are characterized by a lack of coverage from one or more datasets. Generally,
the three datasets seem comparable over most of the globe, thanks to the use of time-aggregation and
standardization procedures that remove temporal inconsistencies and biases among the series. Focusing
the analysis only on the areas where the three datasets are substantially in agreement (following a linear
regression analysis), five macro-regions were detected as suitable for further investigations according to
the Triple Collocation (TC) technique. Under the hypothesis that certain criteria are met, the TC
analysis allows quantifying the likely random error associated with each model (with regard to the true
status) even in absence of an observation of the "truth"..
The main outcome of the TC analysis further confirms the need of a multi-source approach for a
reliable assessment of soil moisture anomalies over those five regions, given that no model outperforms
the others (in terms of expected error variance) for the entire study domain. Emblematic are the results
over North America, where each model outperforms the others in one sub-region, like the LIS approach
in Eastern U.S., LST in the Southern-Western domain and CCI in Central U.S. Even if no clear insight
on the general patterns of the errors can be provided as outcome of the study, overall, the obtained
results seem suggesting that remote sensing datasets perform better over dry areas and sparsely
monitored areas (e.g., Australia and Southern Africa), whereas the LIS dataset seems more reliable over
NA and EU where dense networks of meteorological ground stations are deployed.
It has been highlighted how some differences among the datasets can also be related to the depth
of the soil layer monitored by each dataset, i.e., the microwave system capturing the skin soil moisture
whereas Lisflood models the full root zone; indeed, even if the use of monthly anomalies allows
minimizing some of the discrepancies and biases, our results confirm that skin soil moisture remains
more reliable for areas where the effects of vegetation coverage are minimal (Das et al., 2015), whereas
hydrological models are more suited for agricultural and densely vegetated regions. However, the three
datasets seems to be overall comparable in terms of average performances, supporting the success of the
adopted homogenization procedures. Some analogies between the obtained results and the ones already
available in the literature have been found, but the inclusion of thermal data into the analysis enlarges
the understanding of the mutual relationship between the different datasets.
The results of this study represent a robust starting point for the development of a global drought
monitoring system based on such anomaly datasets, which can exploit the main findings of the TC
analysis in order to develop a suitable ensemble product over the investigated regions. The error
characterization derived from TC was used to estimate the weighing factors of an ensemble mean
procedure, based on the least squares framework reported in Yilmaz et al. (2012). Currently, an
operational implementation of such ensemble product is foreseen for the GDO system as soon as the
CCI product becomes available in near-real time.
Further analyses are required to be able to extend the test to the areas currently not included in this
study, especially the ones where the three datasets are available but provide inconsistent or contrasting
results. In this context, the analysis of further global datasets may help to unveil the reasons behind such
discrepancies.

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

**Tables**

**Table 1.** Summary of the Pearson correlation coefficient values (average ± standard deviation) observed for all the regions.

| Comparison | ALL | NA | EU | SA | IN | AU |
|---|---|---|---|---|---|---|
| LIS vs. LST | 0.44 ± 0.09 | 0.41 ± 0.08 | 0.39 ± 0.07 | 0.48 ± 0.09 | 0.44 ± 0.07 | 0.50 ± 0.10 |
| LIS vs. CCI | 049 ± 0.10 | 0.47 ± 0.09 | 0.42 ± 0.08 | 0.48 ± 0.10 | 0.48 ± 0.08 | 0.58 ± 0.11 |
| CCI vs. LST | 0.56 ± 0.13 | 0.49 ± 0.14 | 0.37 ± 0.09 | 0.63 ± 0.09 | 0.52 ± 0.10 | 0.68 ± 0.07 |

**Table 2.** Summary of the TC error variance analysis, reporting the spatial average (± standard deviation) values observed over each macro-region.

| Model | ALL | NA | EU | SA | IN | AU |
|---|---|---|---|---|---|---|
| LIS | 0.48 ± 0.13 | 0.42 ± 0.14 | 0.44 ± 0.12 | 0.54 ± 0.11 | 0.49 ± 0.10 | 0.54 ± 0.14 |
| LST | 0.44 ± 0.13 | 0.46 ± 0.15 | 0.56 ± 0.10 | 0.37 ± 0.10 | 0.48 ± 0.09 | 0.38 ± 0.11 |
| CCI | 0.36 ± 0.18 | 0.46 ± 0.16 | 0.54 ± 0.12 | 0.30 ± 0.14 | 0.38 ± 0.16 | 0.17 ± 0.10 |

**Table 3.** Fraction of each macro-area (as percentage) where one model outperforms the other two.

| Model | ALL | NA | EU | SA | IN | AU |
|---|---|---|---|---|---|---|
| LIS | 25.5 | 39.2 | 50.0 | 10.6 | 28.2 | 4.3 |
| LST | 25.7 | 28.8 | 23.1 | 36.0 | 20.3 | 18.6 |
| CCI | 48.8 | 32.0 | 26.9 | 53.4 | 51.5 | 77.1 |

**Figures**

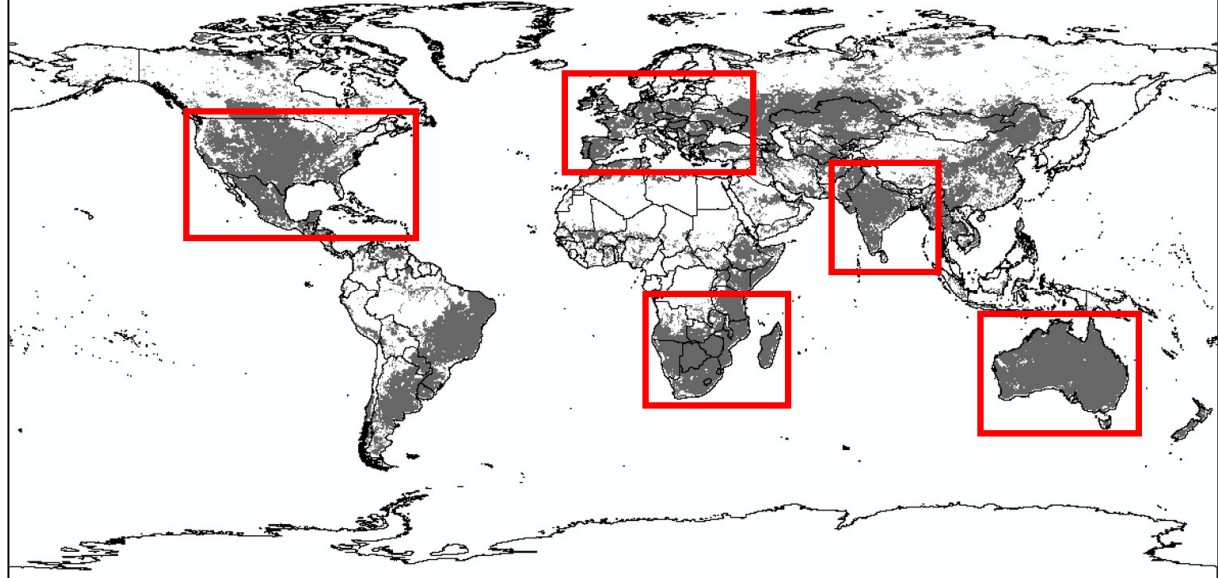


**Fig. 1.** Map of the areas where all the three models are positively significantly linearly correlated (cells
in grey) according to the Student's t-test at $p = 0.05$. The boxes delimitate the macro-regions selected
for the successive analyses.

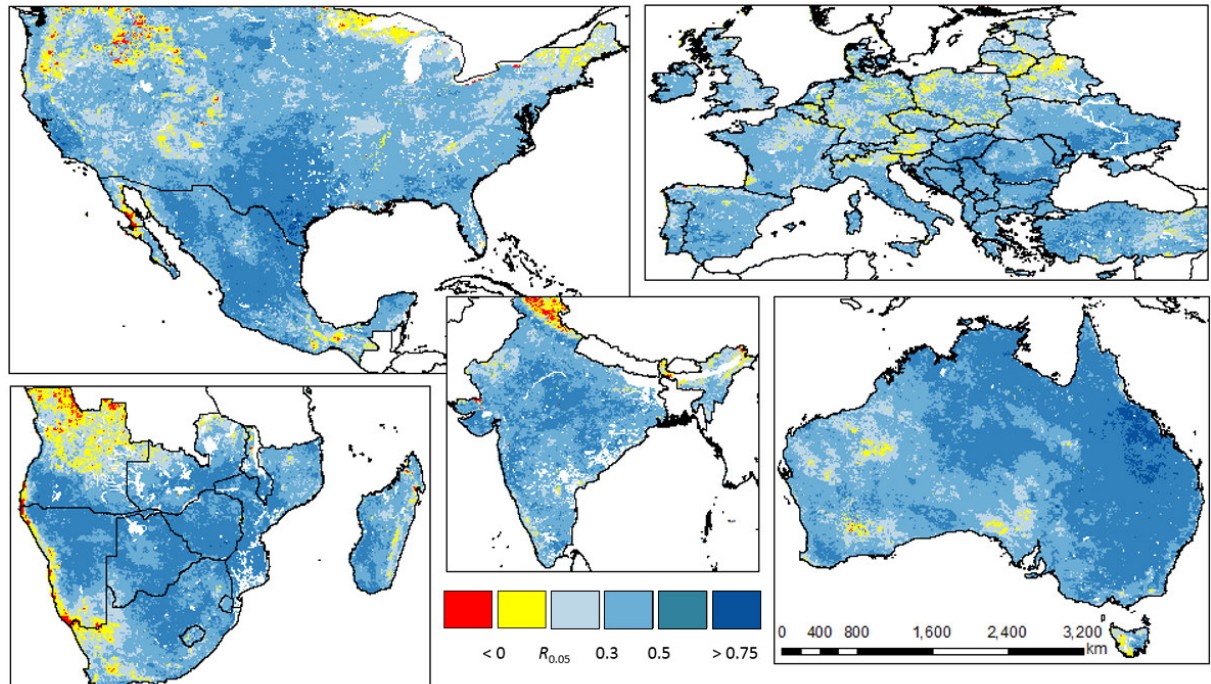

Fig. 2. Spatial distribution of the Pearson correlation coefficient (*R*) between Lisflood soil moisture anomalies (LIS) and land surface temperature anomalies (LST) over the five selected macro-regions. Values in red and yellow are negatively correlated or not significant at $p = 0.05$, respectively.

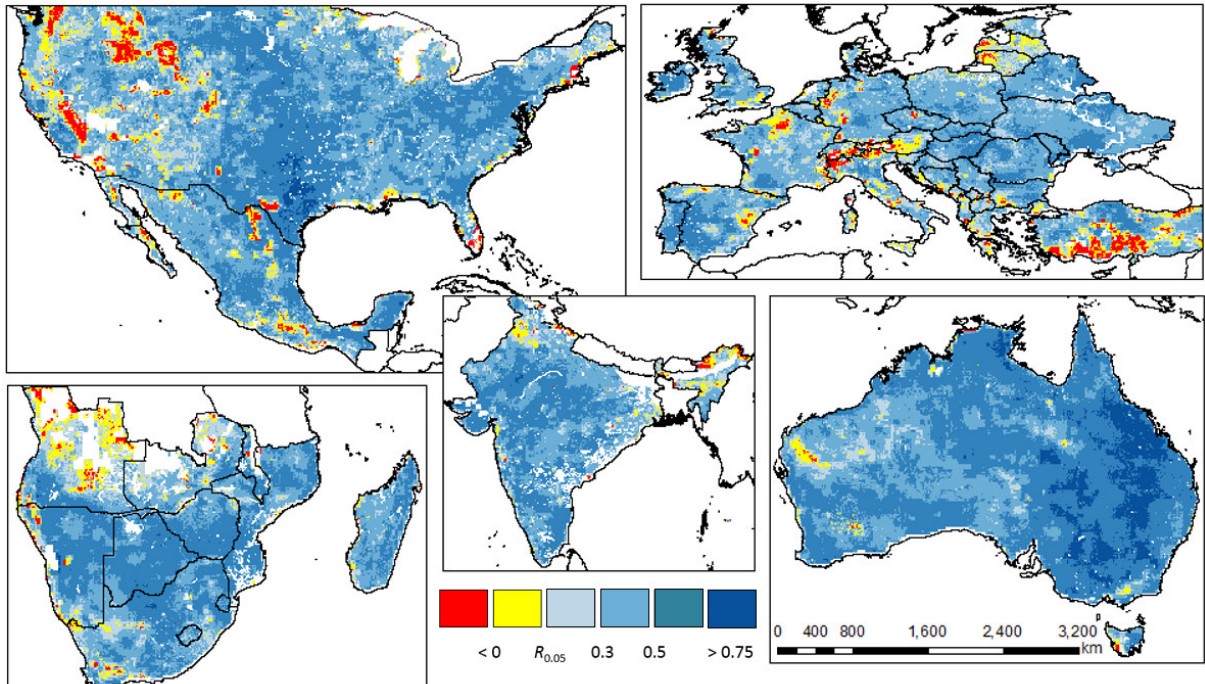


**Fig. 3.** Spatial distribution of the Pearson correlation coefficient ($R$) between Lisflood (LIS) and ESA
Climate Change Initiative (CCI) soil moisture anomalies over the five selected macro-regions. Values in
red and yellow are negatively correlated or not significant at $p = 0.05$, respectively.

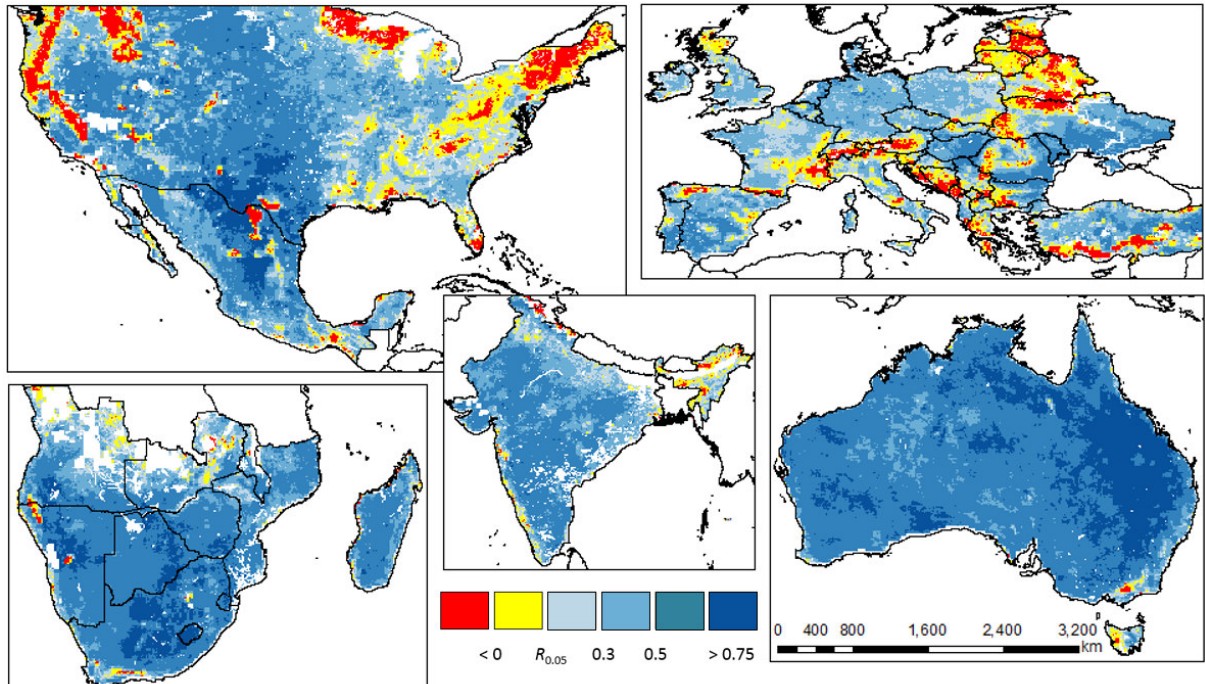


**Fig. 4.** Spatial distribution of the Pearson correlation coefficient (*R*) between ESA Climate Change

Initiative soil moisture anomalies (CCI) and land surface temperature anomalies (LST) over the five

selected macro-regions. Values in red and yellow are negatively correlated or not significant at $p = 0.05$,

respectively.

640

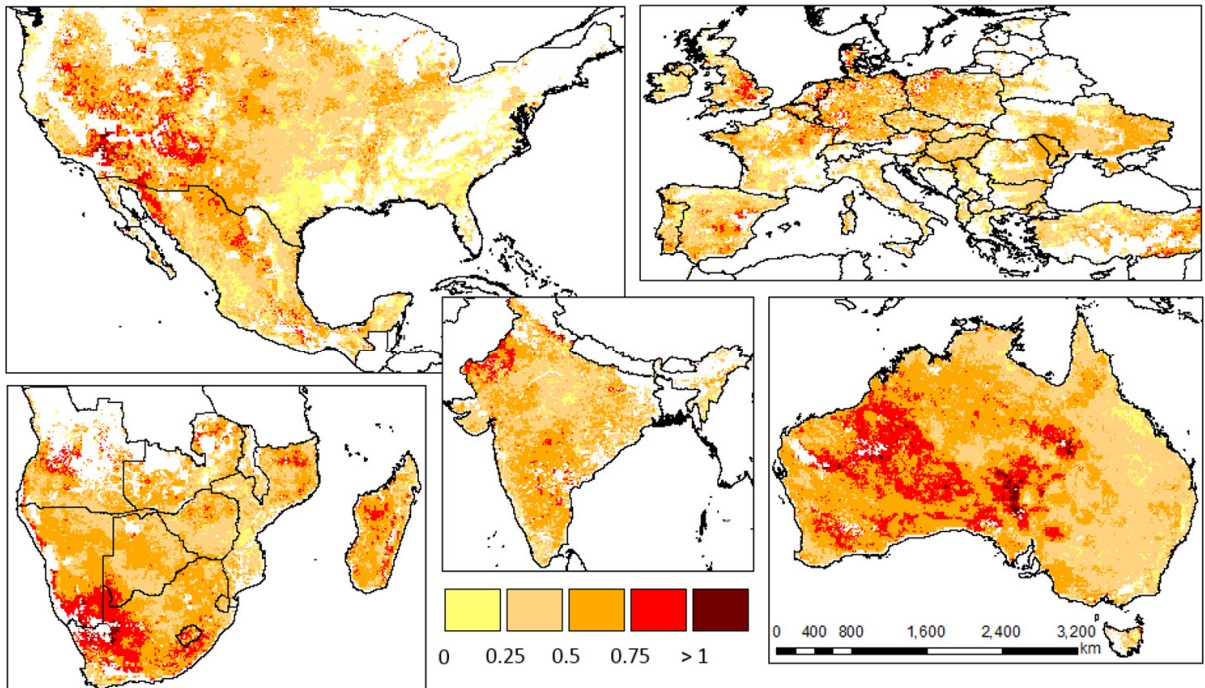

641

**Fig. 5.** Spatial distribution of the error variance for the Lisflood (LIS) dataset over the five selected

macro-regions.

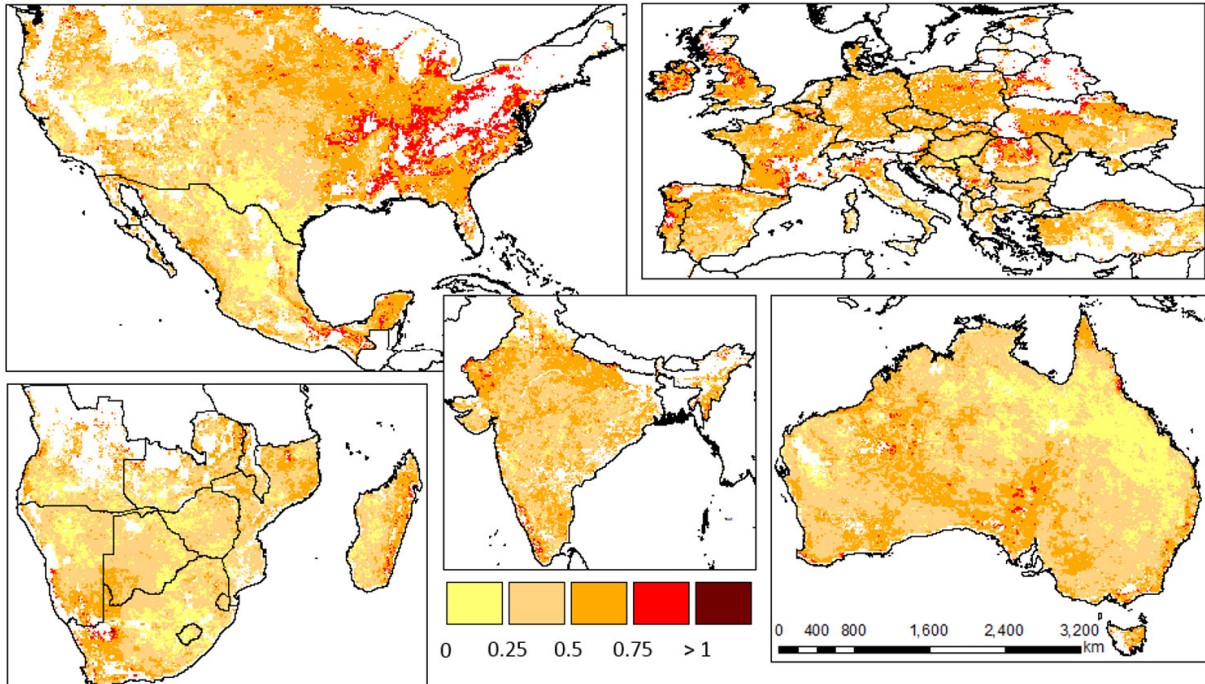

**Fig. 6.** Spatial distribution of the error variance for the land surface temperature (LST) dataset over the

five selected macro-regions.

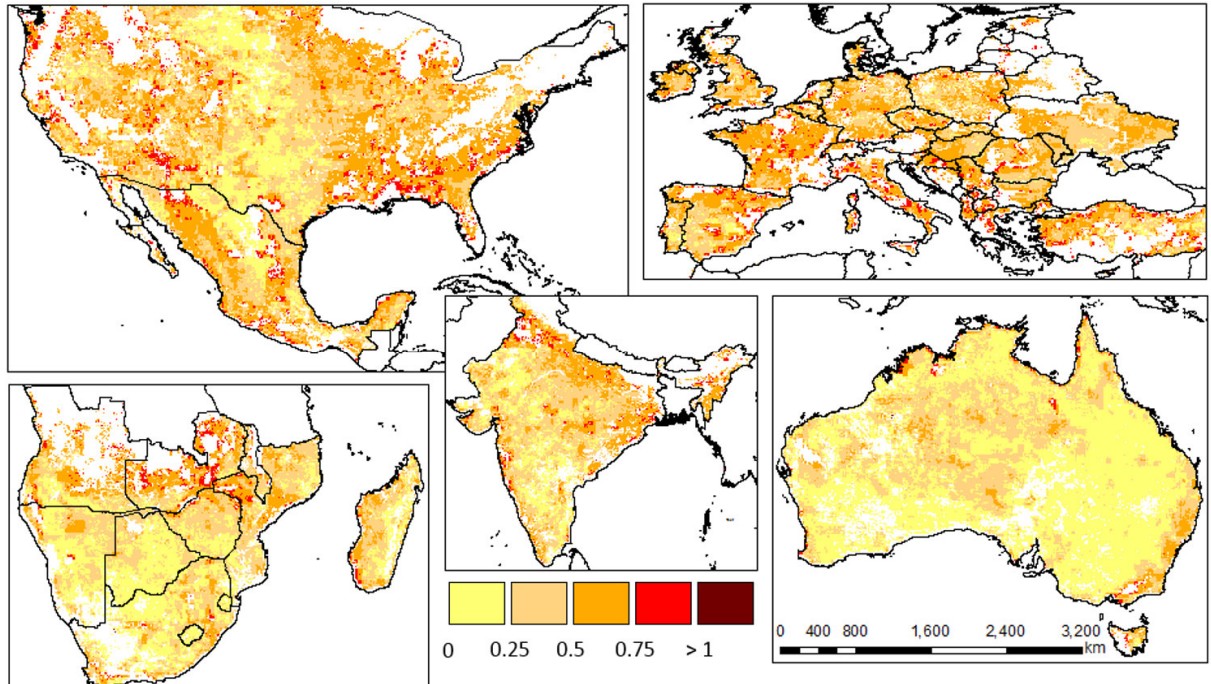


**Fig. 7.** Spatial distribution of the error variance for the ESA Climate Change Initiative (CCI) dataset

over the five selected macro-regions.


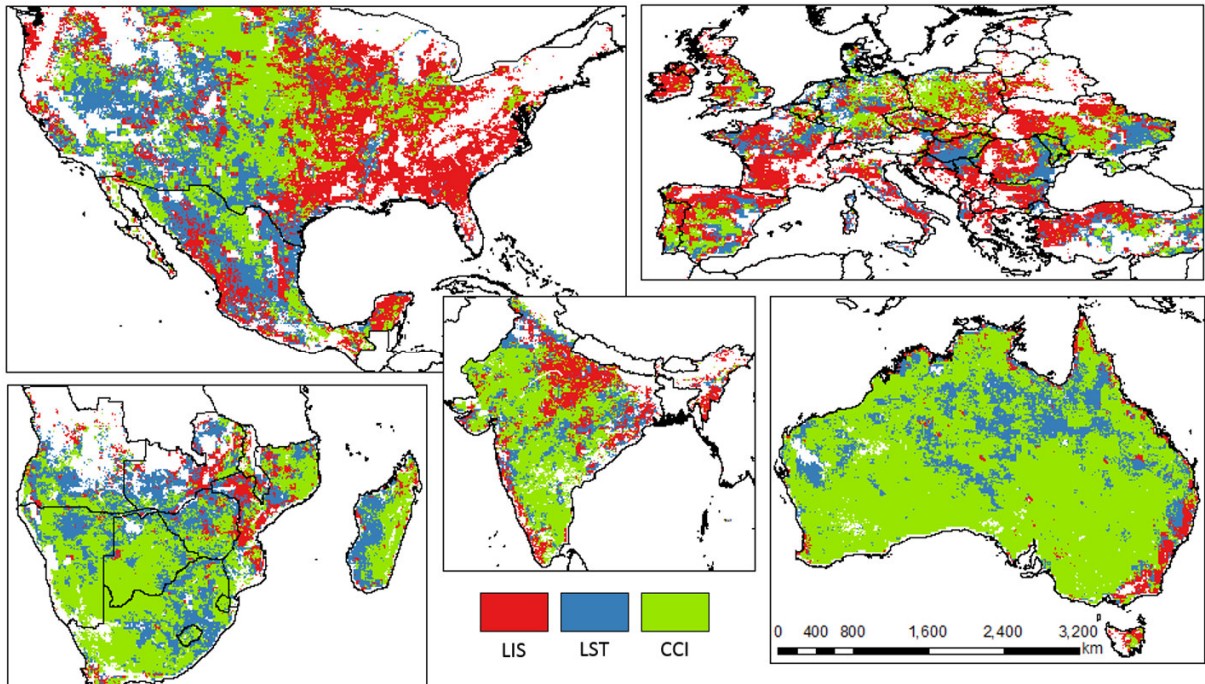


**Fig. 8.** Maps representing the best performing (lowest error variance) dataset for each cell according to
the TC analysis.

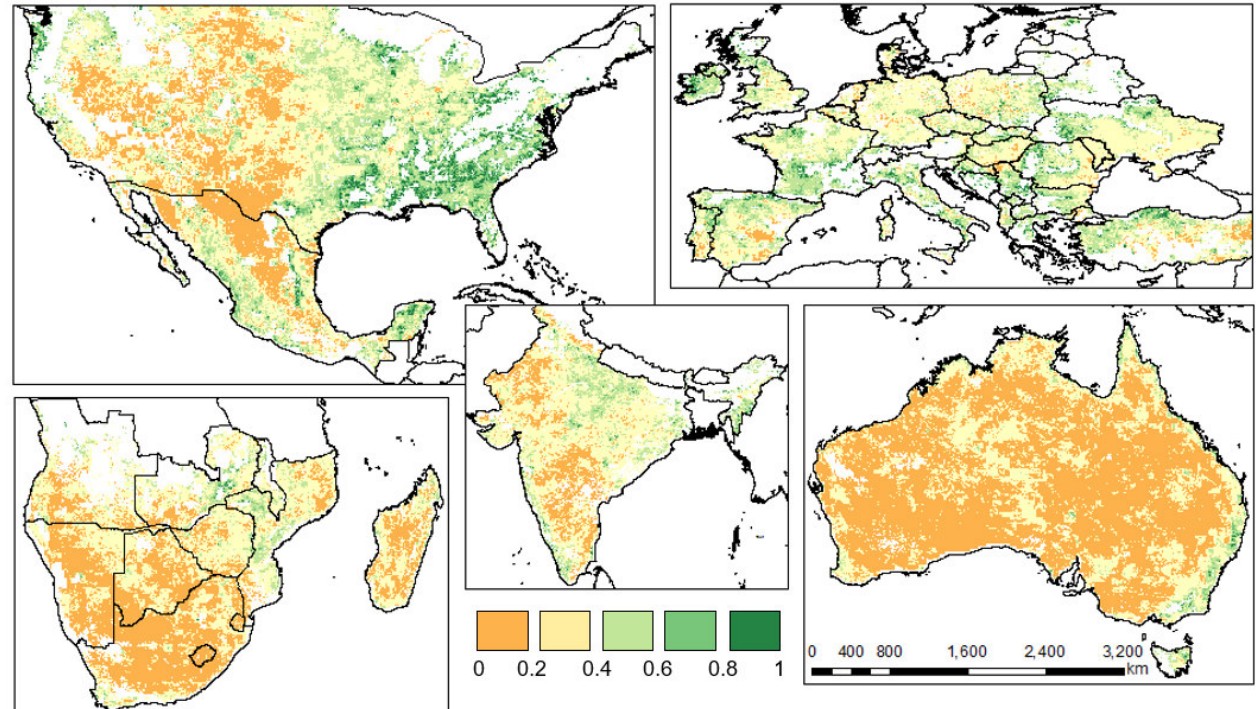


**Fig. 9.** Maps representing the ensemble mean weighting factor for the LIS dataset according to the error

maps derived from the TC analysis.


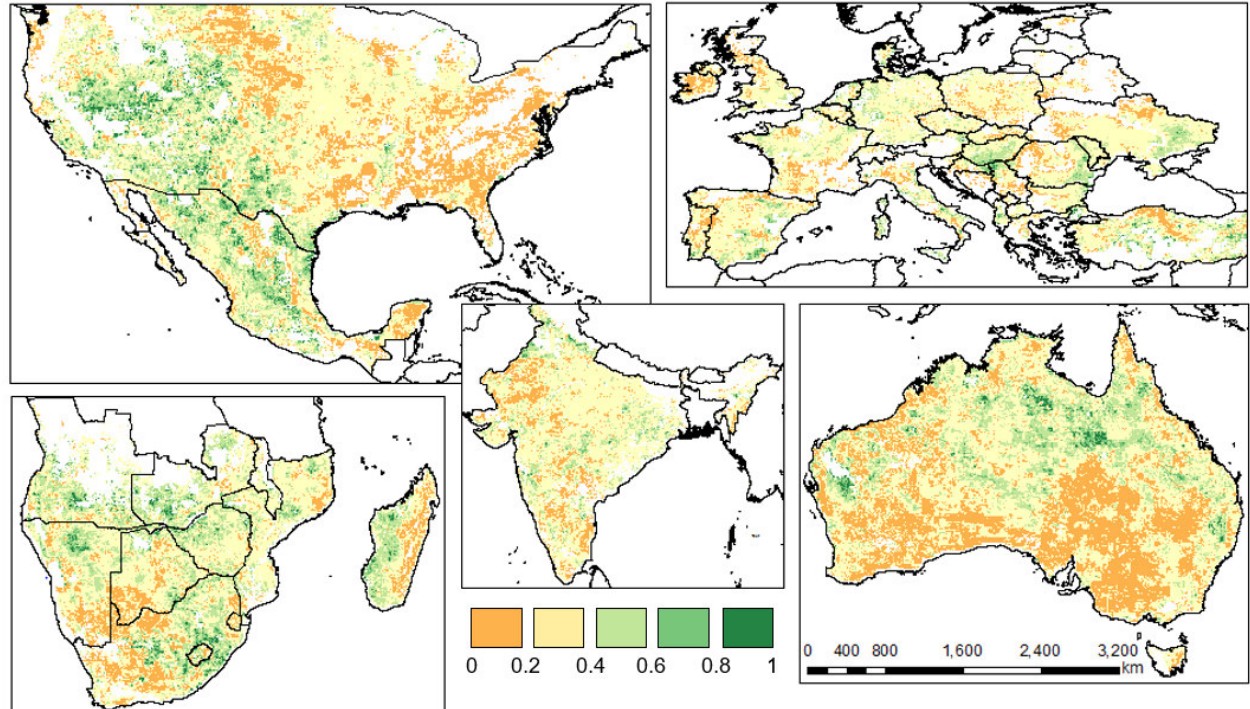


**Fig. 10.** Maps representing the ensemble mean weighting factor for the LST dataset according to the

error maps derived from the TC analysis.



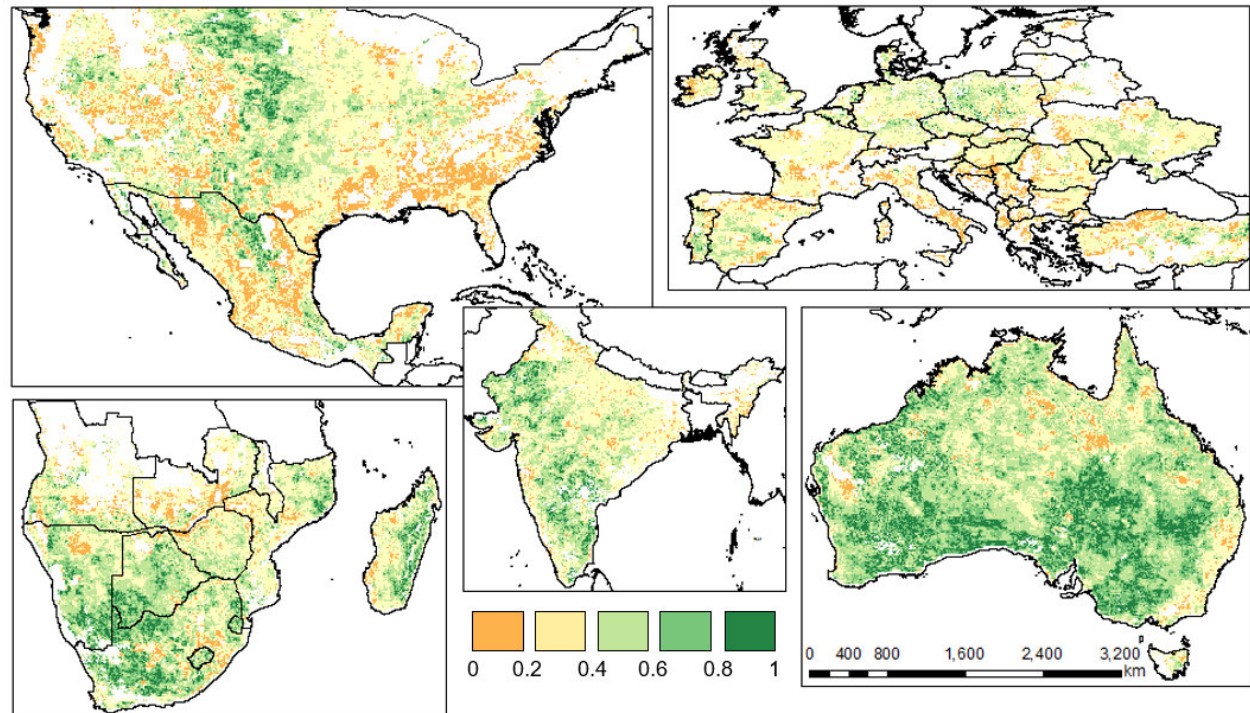


**Fig. 11.** Maps representing the ensemble mean weighting factor for the CCI dataset according to the

error maps derived from the TC analysis.


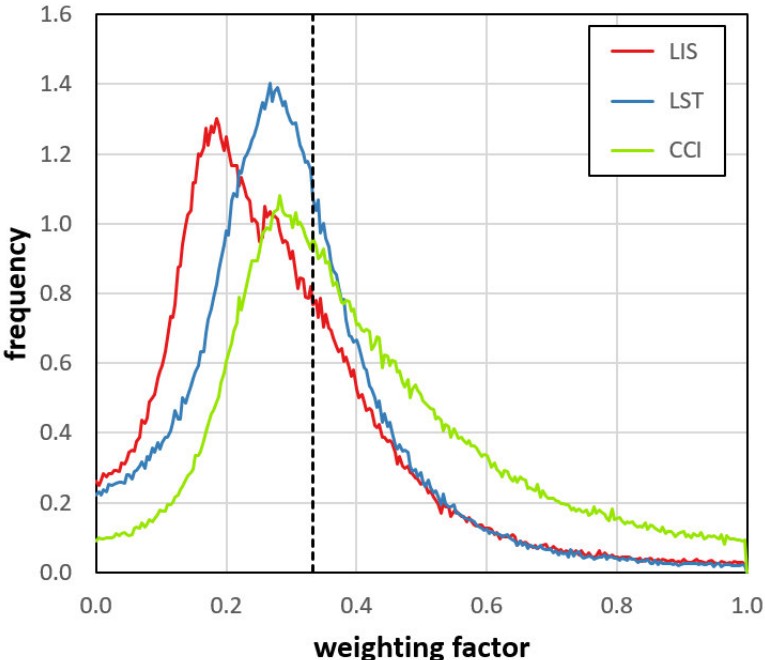


**Fig. 12.** Frequency distribution of the ensemble mean weighting factor for each dataset computed according to the TC analysis. The black dotted line represents the value corresponding to a simple arithmetic average (1/3).



