# Peer review of "Comparing soil moisture anomalies from multiple independent sources over 1 different regions across the globe 2 3 4 Carmelo Cammalleri, Jürgen V. Vogt, Bernard Bisselink, Ad de Roo European Commission, Joint Research Centre (JRC), Ispra, Italy. 5 6 7 Correspondence to: C. Cammalleri, European Commis"

_Hydrology and Earth System Sciences, 2017_

## Referee Comment (RC1) · Anonymous Referee #1 · 8 Jun 2017

General

This study investigates the potential of one model-based and two remotely sensed datasets for their use in an established global drought monitoring system. The study uses soil moisture from the Lisflood model, surface soil moitsure from the microwave-based ESA CCI soil moisture dataset and land Surface temperature from MODIS. Random errors of these datasets are characterised with the triple collocation analysis (TCA), a firmly established method in soil moisture analysis. As such, apart from applying the TCA to a different triple of datasets, the study is not very innovative. However, since the datasets are expected to be used in an operational drought monitoring sys-
tem, the results of this study are expected to have a large practical impact. This study is performed in a scientifically sound and clearly structured way and provide some interesting insights in the skill of the datasets considered. I therefore recommend its publication after addressing the following issues.

Major comments

1. The analyses in this study are based on monthly anomalies. Although theoretically it is feasible to do so, I wonder what the practical relevance of these results are. All datasets used are also available at a daily time step and already now the European Drought Observatory works with ten-day periods (dekades). Hence, also the error structures should be known at these time scales .

2. I was expecting a more thorough analysis on the consequences of using three datasets that represent very different layer depths, i.e.:

* ESA CCI soil moisture represents the upper ∼2 cm, * LST represents the skin temperature, which is driven both by surface soil moisture (influencing bare soil temperature) and root zone soil moisture (impacting vegetation canopy temperature) * LIS represents root zone soil moisture (but layer depth is not provided in the manuscript).

Typically, deeper and thicker layers have lower random errors than observations of the surface layer, but this also depends on the time scale you're looking at (e.g. daily observations typically have larger random errors than monthly averages). Is there a way you can test the impact of using such different layer depths, e.g. by using the surface layer of LISFLOOD?

3. Some important information is missing (or not clearly provided) which is needed for a correct interpretation of the results: in which units are the TCA results expressed? Is it the fractional RMSE? If not, are the errors of each dataset expressed in its own data space or in a common data space provided by one of the models?

Minor comments

Line 16: why do you use the word proxy here? Only LST can be considered a proxy of soil moisture, while both LIS and ESA CCI are real estimates of soil moisture.

Line 20: The official name is ESA CCI Soil Moisture or ESA CCI SM

line 43: When agricultural droughts start affecting human welfare, people commonly use the term socioeconomic drought

Lines 47/48: include references or URLs to these drought monitoring systems

Lines 51/52: these citations refer to the formulation of drought indices rather than soil moisture

Lines 83-97: please provide references to the various datasets discussed here (MODIS LST, ESA CCI SM, LISFLOOD)

Regarding the skill of LST-based soil moisture versus ESA CCI SM the authors should also refer to the ALEXI-based work of Fang et al. 2016 (http://www.sciencedirect.com/science/article/pii/S0303243415300404)

Line 93: ESA CCI SM will soon be updated in NRT in the framework of Copernicus Climate Change Services (http://www.sciencedirect.com/science/article/pii/S0303243415300404)

Line 97: The use of the term "models" is confusing ans only applies to Lisflood. Replace "models" with "datasets" or "products", also throughout the rest of the manuscript.

Line 120: No definition is given of the root zone soil moisture simulated with Lisflood. What soil column is sampled?

line 128: I don't see how Pearson R would give information about the slope and biases between two datasets. Do you confuse it with the "regression function" between the datasets?

Line 130: to my knowledge the correct name for this test is Student's t-test

[Figure]

Line 148: spelling error: where -> were

Line 168: it is unclear why you don't use the surface layer, which is much closer to the other two data sets. I suggest to repeat the TCA with the surface layer as well to see what impact is on the estimated errors of all datasets.

Line 204: It may be worth checking whether using Microwave-based LSTs would lead to similar results (See Holmes et al., 2009; http://onlinelibrary.wiley.com/doi/10.1029/2008JD010257)

Line 225: provide version number of "current" version.

Line 233: Please check Terms and Conditions (http://www.esa-soilmoisture-cci.org/dataregistration/terms-and-conditions) for a correct citation of the data.

Line 236-237: Only for the integration of SSM/I into the merged products the soil moisture signal is decomposed into seasonality and anomalies (Liu et al. (2012))

Line 253: it is not clear whether the linear correlation is computed from the original signal or from the anomalies. Since your TCA implementation is based on the anomalies, also the correlation computation should be based on these.

Line 282-283: Is the stronger correlation between CCI and LST not expected, as they represent more closely related soil layers? This is something you could test by including also the surface layer of Lisflood in your analysis.

Line 307: On one hand -> On the one hand

Line 331: the results of this manuscript cannot be directly compared to those of Pierdicca et al., since the latter applies the TCA to daily observations.

Line 397: For what is skin soil moisture more reliable? Do you mean the estimates themselves? The estimation of soil moisture from microwave remote sensing may have large uncertainties over dry areas (e.g. Hahn et al., 2017: http://ieeexplore.ieee.org/document/7815274/)

Line 406: There are several studies that combine various datasets with different error characteristics, e.g. Liu et al., 2011 (http://www.hydrol-earth-syst-sci.net/15/425/2011/hess-15-425-2011.html); Beck et al., 2017 (http://www.hydrol-earth-syst-sci.net/21/589/2017/hess-21-589-2017.html); Yilmaz et al; 2012 (http://onlinelibrary.wiley.com/doi/10.1029/2011WR011682/full). Is there something that you can learn from these studies for your application?

───────────────────

---

## Referee Comment (RC2) · Anonymous Referee #2 · 12 Jun 2017

In the paper titled "Comparing soil moisture anomalies from multiple independent sources over different regions across the globe" authors have investigated the anomaly components of three products and then compared the errors of the standardized products over five different regions. Overall, the manuscript is written well and appropriate to the journal Hydrology and Earth System Sciences. However, there are some parts still need improvement:

- There are other soil moisture inter-comparison studies performed before at global scale, including the ones that have already implemented TCA. First of all authors should explicitly justify the need of anomaly comparisons at large scales (i.e., not per-

formed before over these locations using anomalies?). It is not all that clear what additional benefit do readers get from this study compared to the earlier studies (i.e., the analysis performed here are not performed before?). What is the new thing? Datasets? Locations? Anomaly components investigated instead of entire datasets? The information given in the introductions should better be tied with the overall goal.

- What is the vertical support of the soil moisture product comparison performed here considering modeled soil moisture reflect root-zone while the MODIS LST and ESA CCI soil moisture products reflect the top couple cm depths. How does it relate to the overall framework of the study (agricultural drought monitoring while root-zone soil moisture lies at the hearth of such analyses in general)? Surface skin soil moisture is a good indicator for agricultural drought??

- Percentage error variance information is not all that helpful, perhaps actual standard deviations (volumetric error for the model and satellite soil moisture products and K for the LST product) would be more helpful (i.e., how do these errors relate to specific mission goals of 4%). Or at least authors should justify why presentation of standardized error variance is a better thing to do compared to actual error variance.

- Error variance comparisons of three datasets in space is done, but it would be helpful if more is given.. For example, specific pattern between the error variance and vegetation/precipitation/elevation distributions? Any one better under such and such conditions (instead of only locations)? Why better? A dedicated paragraph would be very helpful.

- Combination of different datasets is spelled out in the introduction (final paragraph where the goal of the study is stated) but not performed (it gives the impression that this study will merge different products; perhaps it should have given all the necessary inputs are available including the error sources of each product).

- Some background discussion about the TCA dataset requirements/assumptions (e.g., length? See Zwieback et al, 2012, doi:10.5194/npg-19-69-2012).

L98: "two folds" L102: "to develop a suitable combination procedure for a near-real time detection of the occurrence of ecosystem drought events".. More specifics. How this will be performed? Using TCA errors to calculate the weights in a merging algorithm? L112: revise "in order to make directly comparable the different datasets"

———————————————

---

## Author Comment (AC1) · 19 Jun 2017

We would like to thank to the anonymous reviewer for the thoughtful comments. Here we provide a brief response to the major comments to highlight the edits that will be made on the manuscript to address them.

1) The use of time-aggregated data is a common practice in drought analyses in order to ensure the statistical robustness of the computed anomalies. Daily values are often too noisy to allow for a robust statistical analysis. The use of a monthly temporal scale is quite common in the drought literature, and it has been preferred over the ten-day scale (adopted in EDO) for operational/technical reasons. Indeed, the monthly time

scale will be the one implemented in the GDO system as a first approach. Higher temporal resolution will be tested and implemented in the future. We will clarify this point in the revised version.

2) Additionally, the use of monthly aggregated data ensures minimizing the discrepancies between the three datasets related to various layer depths (as the reviewer correctly highlighted in comment 2). We will discuss more thoughtful the implications of such choice in the new version of the manuscript.

3) Since standardized anomalies are used (dimensionless, expressed as multiples of the standard deviation), the units of the errors are expressed in this unit as well. The standardization procedure also allows having all the three datasets in the same range of variability (zero mean, unitary standard deviation), which removes the need of a reference data space. Additionally, as briefly stated in the "Methods" section, we adopted the TCA "covariance notation", which does not require a common (arbitrary) reference dataset. We will further clarify this point in the revised text.

---

## Author Comment (AC2) · 19 Jun 2017

We appreciate the detailed analysis provided by the anonymous reviewer, and we would like to provide some brief reply in order to detail the edits that will be made to the manuscript in order to accommodate the main concerns.

1) The motivation of the study will be better explained in the revised version of the manuscript. In summary, most of the past TCA studies were focused on soil moisture data rather than anomalies, and they often focus on modelled/microwave datasets only. Hence, the data analyzed here are unique in terms of both variable investigated and datasets adopted. Also, the operational focus of this study (implementation in GDO)

requires an analysis of the datasets that can be actually used in the monitoring system.

2) Similarly to the reply to reviewer #1, the use of both standardized anomalies and monthly data should minimize the discrepancies among the datasets. This point will be further clarified and discussed in the revised manuscript.

3) Actual error variance (i.e., in terms of volumetric water content) cannot be reported in this study, since standardized variables (normalized z-scores) are investigated. The use of standardized quantities is justified by the needs of drought monitoring, as it will be further clarified in the new text.

4) We investigated possible relationships between error patterns and climatic/vegetation distribution, but we have found only few weak connections. We will discuss this further in the revised version of the text.

5) The ultimate goal of this study is to provide information on spatial patterns of likely errors of each product that can be used to obtain a reliable ensemble product for an operational global drought monitoring system. Indeed, the outputs of the presented analysis are used in the operation of GDO, but the ensemble product itself is not discussed here. We will clarify that the ensemble product is not the final goal of the reported study, but that the goal of the research is to spatially characterize the errors to be used in the operational system.

---

## Author Response (AR1)

**Revision of the paper HESS-2017-196**

We would like to thank to both the anonymous reviewers for the thoughtful comments provided to our manuscript. A point-to-point reply to major and minor comments is reported below, as well as indication of the changes made to the text to address such comments.

**Reply to Referee #1 Comments**

General

This study investigates the potential of one model-based and two remotely sensed datasets for their use in an established global drought monitoring system. The study uses soil moisture from the Lisflood model, surface soil moisture from the microwave-based ESA CCI soil moisture dataset and land Surface temperature from MODIS. Random errors of these datasets are characterized with the triple collocation analysis (TCA), a firmly established method in soil moisture analysis. As such, apart from applying the TCA to a different triple of datasets, the study is not very innovative. However, since the datasets are expected to be used in an operational drought monitoring system, the results of this study are expected to have a large practical impact. This study is performed in a scientifically sound and clearly structured way and provide some interesting insights in the skill of the datasets considered. I therefore recommend its publication after addressing the following issues.

Thanks for your comment. Regarding the innovative aspect of the work, we agree that the application of triple collocation is nothing new in the scientific literature of soil moisture; however, there are very few studies analyzing modelled, microwave and thermal data, and none of them are focusing on the study areas reported here. Additionally, the use of standardized anomalies in TCA is uncommon in the literature even if it is a key aspect in drought studies. As you stated, the final goal of the study is to provide insight on the spatial distribution of the errors in our global drought observatory, hence the analysis of these three specific datasets (previously uninvestigated in TCA literature) and anomaly values is a novelty and a key requirement of our study. We clarified these novelties of the study in the new introduction section of the paper.

Major comments

1. The analyses in this study are based on monthly anomalies. Although theoretically it is feasible to do so, I wonder what the practical relevance of these results are. All datasets used are also available at a daily time step and already now the European Drought Observatory works with ten-day periods (dekades). Hence, also the error structures should be known at these time scales.

The use of time-aggregated data is a common practice in drought analyses in order to ensure the statistical robustness of the computed anomalies. Daily values are often too noisy to allow for a robust statistical analysis. The use of a monthly temporal scale is quite common in the drought literature, and it has been preferred (at the moment) over the ten-day scale (adopted in EDO) for operational/technical reasons. Indeed, the monthly time scale will be the one implemented in the GDO system as a first approach. Higher temporal resolutions will be tested and implemented in the future. We clarified this point in the revised version of the manuscript.

2. I was expecting a more thorough analysis on the consequences of using three datasets that represent very different layer depths, i.e.:

* ESA CCI soil moisture represents the upper 2 cm, * LST represents the skin temperature, which is driven both by surface soil moisture (influencing bare soil temperature) and root zone soil moisture (impacting vegetation canopy temperature) * LIS represents root zone soil moisture (but layer depth is not provided in the manuscript).

Typically, deeper and thicker layers have lower random errors than observations of the surface layer, but this also depends on the time scale you're looking at (e.g. daily observations typically have larger random errors than monthly averages). Is there a way you can test the impact of using such different layer depths, e.g. by using the surface layer of LISFLOOD?

We agree that the different vertical (as well as horizontal) resolution of the three datasets may lead to some discrepancies in the analysis. For this reason, we elaborated the data in order to try to minimize the possible discrepancies among the datasets. For instance, the use of monthly aggregated data is one of the techniques adopted to ensure minimizing the discrepancies between the three datasets related to the slower response of deeper soil layers (as you highlighted). Also, the use of standardized quantities allows reducing the effects of possible biases between soil moisture modelled at different depths.

We modified the text to highlight further the expedients adopted to try to minimize those issues, as well as highlighting where the different resolutions may lead to a better performance of one dataset over the others.

3. Some important information is missing (or not clearly provided) which is needed for a correct interpretation of the results: in which units are the TCA results expressed? Is it the fractional RMSE? If not, are the errors of each dataset expressed in its own data space or in a common data space provided by one of the models?

We better clarified that the errors are provided as dimensionless quantities (multiple of standard deviation), due to the standardization applied to the three datasets before performing the TCA. The standardization procedure also allows having all the three datasets in the same range of variability (zero mean, unitary standard deviation), which removes the need of a reference data space. Additionally, as briefly stated in the "Methods" section, we adopted the TCA "covariance notation", which does not require a common (arbitrary) reference dataset. We highlighted that information in the revised text.

Minor Comments

Line 16: why do you use the word proxy here? Only LST can be considered a proxy of soil moisture, while both LIS and ESA CCI are real estimates of soil moisture.

We rephrased this sentence.

Line 20: The official name is ESA CCI Soil Moisture or ESA CCI SM.

We reworded this definition of the product.

line 43: When agricultural droughts start affecting human welfare, people commonly use the term socioeconomic drought.

In our opinion, socioeconomic drought is a much larger concept that includes also other economic factors and that is separated from the classical "physical" classification meteorological/hydrological/agricultural. We prefer to leave this sentence as it is.

Lines 47/48: include references or URLs to these drought monitoring systems.

Done.

Lines 51/52: these citations refer to the formulation of drought indices rather than soil moisture.

We clarified that we were referring to soil moisture modelling in the context of drought.

Lines 83-97: please provide references to the various datasets discussed here (MODIS LST, ESA CCI SM, LISFLOOD).

References are reported in the successive sub-sections dedicated to each product. We prefer to leave this part of the introduction easy to read by avoiding further references.

Regarding the skill of LST-based soil moisture versus ESA CCI SM the authors should also refer to the ALEXI-based work of Fang et al. 2016 (http://www.sciencedirect.com/science/article/pii/S0303243415300404).

Thanks. We added some comments on this work in the results section.

Line 93: ESA CCI SM will soon be updated in NRT in the framework of Copernicus Climate Change Services (http://www.sciencedirect.com/science/article/pii/S0303243415300404).

Yes, we added this information to the text.

Line 97: The use of the term "models" is confusing and only applies to Lisflood. Replace "models" with "datasets" or "products", also throughout the rest of the manuscript.

Done.

Line 120: No definition is given of the root zone soil moisture simulated with Lisflood. What soil column is sampled?

Information have been added to sub-section 3.1.

line 128: I don't see how Pearson R would give information about the slope and biases between two datasets. Do you confuse it with the "regression function" between the datasets?

This is true only for the specific case of a regression analyses between two standardized quantities with zero mean and a unitary variance. The text has been amended to clarify this point.

Line 130: to my knowledge the correct name for this test is Student's t-test.

Done.

Line 148: spelling error: where -> were.

Done.

Line 168: it is unclear why you don't use the surface layer, which is much closer to the other two data sets. I suggest to repeat the TCA with the surface layer as well to see what impact is on the estimated errors of all datasets.

We agree that the LIS surface layer is closer to CCI but not necessarily to LST (which depth depends on vegetation coverage). We preferred root zone because it is more relevant for agricultural drought studies. Currently, we are testing the value of extrapolated "root-zone-like" soil moisture from skin CCI values, but this analysis is going beyond the goal of this paper.

Line 204: It may be worth checking whether using Microwave-based LSTs would lead to similar results (See Holmes et al., 2009; http://onlinelibrary.wiley.com/doi/10.1029/2008JD010257).

We are aware of the studies on microwave LST, but we do not think that these are relevant in this specific case study, since they compare quite well with MODIS data under clear sky conditions. The main advance of microwave-based LST dataset is to remove the limitation of thermal data during cloudy days, which is not impactful at monthly time scale. Also, microwave data have, at the moment, a coarser spatial resolution.

Line 225: provide version number of "current" version.

Done.

Line 233: Please check Terms and Conditions (http://www.esa-soilmoisture-cci.org/dataregistration/terms-and-conditions) for a correct citation of the data.

References have been updated.

Line 236-237: Only for the integration of SSM/I into the merged products the soil moisture signal is decomposed into seasonality and anomalies (Liu et al. (2012)).

Yes, you are right. However, this is just a very brief summary of the procedure and we think that the readers can find a very clear and detailed description of the procedure (if they are interested) in the cited paper.

Line 253: it is not clear whether the linear correlation is computed from the original signal or from the anomalies. Since your TCA implementation is based on the anomalies, also the correlation computation should be based on these.

Yes, it is computed on the anomalies. We clarified this in the text.

Line 282-283: Is the stronger correlation between CCI and LST not expected, as they represent more closely related soil layers? This is something you could test by including also the surface layer of Lisflood in your analysis.

In the new version of the text we added more considerations related to the explored vertical depth of each dataset. Yes, indeed it is expected that LST performs more closely to CCI in low-coverage areas (like Australia) and more closely to LIS over more vegetated areas.

Line 307: On one hand -> On the one hand.

Done.

Line 331: the results of this manuscript cannot be directly compared to those of Pierdicca et al., since the latter applies the TCA to daily observations.

Yes, this is true. This is the reason why we highlighted only some qualitative analogies. We clarified this difference in the two studies in the new version of the text.

Line 397: For what is skin soil moisture more reliable? Do you mean the estimates themselves? The estimation of soil moisture from microwave remote sensing may have large uncertainties over dry areas (e.g. Hahn et al., 2017: http://ieeexplore.ieee.org/document/7815274/).

We agree that under some circumstances microwave products can have large uncertainties over dry sandy soil. We rephrased the sentence to clarify that the presence of vegetation generally tends to reduce the reliability of microwave estimates.

Line 406: There are several studies that combine various datasets with different error characteristics, e.g. Liu et al., 2011 (http://www.hydrol-earth-syst-sci.net/15/425/2011/hess-15-425-2011.html); Beck et al., 2017 (http://www.hydrol-earth-syst-sci.net/21/589/2017/hess-21-589-2017.html); Yilmaz et al; 2012 (http://onlinelibrary.wiley.com/doi/10.1029/2011WR011682/full). Is there something that you can learn from these studies for your application?

We cited most of the reported authors (even if not specifically these papers) in out text already. The majority of these (and other) merging procedures are loosely based on a weighted average approach, with weights derived from the error analysis procedure. Particularly, the approach proposed in Yilmaz et al. (2012) is the one that we are currently implementing in our operational system, of which this study is the error characterization step. We highlighted this in the final section of the new version of the manuscript and we added some maps showing the actual spatial distribution and frequency of the weighting factors, as also suggested by rev. #2.

**Reply to Referee #2 Comments**

In the paper titled "Comparing soil moisture anomalies from multiple independent sources over different regions across the globe" authors have investigated the anomaly components of three products and then compared the errors of the standardized products over five different regions. Overall, the manuscript is written well and appropriate to the journal Hydrology and Earth System Sciences. However, there are some parts still need improvement:

- There are other soil moisture inter-comparison studies performed before at global scale, including the ones that have already implemented TCA. First of all authors should explicitly justify the need of anomaly comparisons at large scales (i.e., not per formed before over these locations using anomalies?). It is not all that clear what additional benefit do readers get from this study compared to the earlier studies (i.e., the analysis performed here are not performed before?). What is the new thing? Datasets? Locations? Anomaly components investigated instead of entire datasets? The information given in the introductions should better be tied with the overall goal.

We modified most of the final part of the introduction in order to highlight the novelty and the motivation of this study, as also requested by Rev. #1. In summary, most of the past TC studies were focused on soil moisture data rather than anomalies, with the latter being a key variable for drought monitoring that can behave quite differently from soil moisture itself. The inclusion of thermal remote sensing datasets is quite rare in the TC literature, and no studies on global scale are available to our knowledge. It follows that, even if the methodology adopted can be considered "standard", the data analyzed here are unique in terms of both variable investigated and datasets adopted. Finally, the operational focus of this study (error analysis to be used in a future implementation in a near-real time monitoring system) requires analyzing specifically those datasets that can be actually used in the GDO monitoring system.

What is the vertical support of the soil moisture product comparison performed here considering modeled soil moisture reflect root-zone while the MODIS LST and ESA CCI soil moisture products reflect the top couple cm depths. How does it relate to the overall framework of the study (agricultural drought monitoring while root-zone soil moisture lies at the hearth of such analyses in general)? Surface skin soil moisture is a good indicator for agricultural drought?

As replied to the second major comment of rev. #1, we adopted some pre-processing procedures in order to minimize likely discrepancies between datasets with different vertical resolution. In this context, the choice of a monthly-aggregation period aimed at removing the time shift between data referring to different soil depths, whereas standardized anomalies removed biases among the three datasets. The obtained results seem to support the idea that these expedites were able to make the datasets comparable overall.

These considerations were further highlighted in the new version of the manuscript.

Percentage error variance information is not all that helpful, perhaps actual standard deviations (volumetric error for the model and satellite soil moisture products and K for the LST product) would be more helpful (i.e., how do these errors relate to specific mission goals of 4%). Or at least authors should justify why presentation of standardized error variance is a better thing to do compared to actual error variance.

Actual error variance (i.e., in terms of volumetric water content or degree) cannot be reported in this study, since standardized variables (normalized z-scores) are investigated for the above mentioned reasons. The use of standardized quantities is justified by the needs of drought monitoring, as clarified in the new version of the manuscript.

Error variance comparisons of three datasets in space is done, but it would be helpful if more is given. For example, specific pattern between the error variance and vegetation/precipitation/elevation distributions? Any one better under such and such conditions (instead of only locations)? Why better? A dedicated paragraph would be very helpful.

We agree that it would be helpful to be able to provide more insight on the spatial patterns of the errors. However, the obtained results do not provide clear evidence of the suggested relations, behind some general behaviors (LIS perform better over well-monitored agricultural areas whereas CCI perform better over remote dry areas). We highlighted the inability to infer further on spatial patterns on the new version of the discussion and conclusion sections.

Combination of different datasets is spelled out in the introduction (final paragraph where the goal of the study is stated) but not performed (it gives the impression that this study will merge different products; perhaps it should have given all the necessary inputs are available including the error sources of each product).

We agree that this sentence may be misleading, and we rephrased the text to clarify this point. Even if the final goal of our project is to provide an operational ensemble product for drought monitoring, the goal of this specific study is "limited" to the characterization of spatial errors for the three selected datasets. The output of this error analysis is used to obtain statistical robust weighting factors for a reliable ensemble product to be used in GDO, and maps of these weighting factors have been added to the new version of the manuscript. Indeed, a first version of this product is already available in a prototype form in GDO, but a full implementation of the ensemble (based on the outcome of this study) is subordinate to the future availability of CCI in near-real time.

In the new version of the manuscript we clarified that the ensemble product is not the final goal of the reported study, but that the goal of the research is to spatially characterize the errors to be used in the future within the operational system. However, more details on the weighting factor have been added to partially fulfil your request.

Some background discussion about the TCA dataset requirements/assumptions (e.g., length? See Zwieback et al, 2012, doi:10.5194/npg-19-69-2012).

We added these considerations to the methods section, as well as some relevant references.

L98: "two folds" L102: "to develop a suitable combination procedure for a near-real time detection of the occurrence of ecosystem drought events".. More specifics. How this will be performed? Using TCA errors to calculate the weights in a merging algorithm?

Yes. As now clarified in the text, the merging procedure will be based on a weighted average with weights derived from the TC errors. Details on the spatial distribution of the weighting factors have been added; however, the implementation of such ensemble product is not yet fully developed in the operational GDO system.

[revised manuscript text omitted]